# R17C Mutation in Photoreceptor Disc-Specific Protein, PRCD, Results in Additional Lipidation Altering Protein Stability and Subcellular Localization

**DOI:** 10.3390/ijms231810802

**Published:** 2022-09-16

**Authors:** Boyden Myers, Emily R. Sechrest, Gabrielle Hamner, Sree I. Motipally, Joseph Murphy, Saravanan Kolandaivelu

**Affiliations:** 1Department of Ophthalmology and Visual Sciences, Eye Institute, One Medical Center Drive, West Virginia University, Morgantown, WV 26506, USA; 2Department of Neurosciences, One Medical Center Drive, West Virginia University, Morgantown, WV 26506, USA; 3Department of Biochemistry and Molecular Medicine, One Medical Center Drive, West Virginia University, Morgantown, WV 26506, USA

**Keywords:** PRCD, retinitis pigmentosa, outer segment, inner segment, polybasic region, Acyl-resin assisted capture (Acyl-RAC), subretinal injection

## Abstract

Progressive rod-cone degeneration (PRCD) is a photoreceptor outer segment (OS) disc-specific protein essential for maintaining OS structures while contributing to rhodopsin packaging densities and distribution in disc membranes. Previously, we showed PRCD undergoing palmitoylation at the sole cysteine (Cys2), where a mutation linked with retinitis pigmentosa (RP) in humans and dogs demonstrates the importance of palmitoylation for protein stability and trafficking to the OS. We demonstrate a mutation, in the polybasic region (PBR) of PRCD (Arg17Cys) linked with RP where an additional lipidation is observed through acyl-RAC. Immunolocalization of transiently expressed R17C in hRPE1 cells depicts similar characteristics to wild-type PRCD; however, a double mutant lacking endogenous palmitoylation at Cys2Tyr with Arg17Cys is comparable to the C2Y protein as both aggregate, mislocalized to the subcellular compartments within the cytoplasm. Subretinal injection of PRCD mutant constructs followed by electroporation in murine retina exhibit mislocalization in the inner segment. Despite being additionally lipidated and demonstrating strong membrane association, the mutation in the PBR affects protein stability and localization to the OS. Acylation within the PBR alone neither compensates for protein stability nor trafficking, revealing defects in the PBR likely lead to dysregulation of PRCD protein associated with blinding diseases.

## 1. Introduction

Retinitis pigmentosa (RP) is the name of a group of inherited blinding diseases characterized by progressive retinal degeneration initially affecting rod photoreceptors, followed by cone photoreceptor death, and ultimately complete loss of vision [1,2,3]. To date, over 90 genes are associated with RP, including the photoreceptor outer segment (OS) disc-specific protein “progressive rod-cone degeneration” (PRCD) [3,4,5,6]. The Prcd gene is expressed exclusively in retinal photoreceptor cells and encodes a small, 54 amino acid (aa) protein in humans and dogs, and a 53 aa protein in mice. In photoreceptors, PRCD is synthesized in the IS, trafficked to the OS, and strongly associated with the disc membranes. Although the precise role of PRCD in the disc membranes remains unclear, our lab has recently demonstrated that PRCD plays a crucial role in regulating the packaging of rhodopsin into OS disc membranes [7,8,9]. Even though PRCD is localized in the OS and interacts with rhodopsin, it does not appear to have a role in phototransduction [8,10,11]. Based on this information, it is clear that PRCD plays an important role in the maintenance of photoreceptor OS structure, a process that when disrupted can result in defects in visual function [12].

As stated, PRCD has been shown to be associated with RP in humans and dogs, causing late-onset degeneration of retinal photoreceptor cells [5]. Six mutations in PRCD have been linked to RP, with the most common mutation being a cysteine to tyrosine mutation at the second amino acid (C2Y) found in over 30 dog breeds and in humans [5,8,12,13,14,15]. Mutation of the sole cysteine leads to loss of palmitoylation in PRCD resulting in severe loss of protein stability and mislocalization to the photoreceptor inner segment (IS) [12]. Although mutations in PRCD do ultimately lead to slow and progressive photoreceptor degeneration, one interesting hallmark of PRCD-associated disease is the heterogeneity that has been observed between different dog breeds and mouse strains in regard to time of disease onset and progression of the disease [8,9,11]. Another mutation in PRCD which results in the substitution of a cysteine for arginine at the 17th amino acid (R17C) is considered to be an ethnic-specific polymorphism observed in a small population of people from India and Pakistan [5]. Although this polymorphism is considered to be non-disease-causing, a patient of Indian origin with a heterozygous R17C mutation in PRCD also presents with RP [5]. Based on the Global variome shared LOVD database accessed on 7 September 2022 (https://databases.lovd.nl/shared/variants/0000796901#00016771), there are four reported entries, one of which was a homozygous mutation reported in 2014 that reveals, by molecular screening using the next generation sequencing (NGS). The patient with retinitis pigmentosa discloses a homozygous mutation of PRCD-R17C with the RP phenotype [16].

Interestingly, the R17C mutation in PRCD protein located within the polybasic region (PBR) consists of three arginine residues following the transmembrane helix. Previous studies demonstrate that the PBR is critical for many cellular functions, such as in small GTPase RAS and Rac1 where the PBR region containing dual lipid modifications is essential for strong membrane binding [17,18,19]. In contrast to other known mutations in this region of PRCD, such as R18X and R22X that result in a premature stop codon, elucidating how an R17C mutation in PRCD could affect protein function or localization could help to provide vital insight into a more specific role for this small protein in photoreceptors. In the current study, we sought to characterize the localization, acylation status, and overall stability of PRCD containing an R17C mutation. We observed in cultured hRPE1 cells transfected with PRCD-R17C that there is a deficiency occurring in the stability of protein, as well as the possibility for an additional lipidation site at Cys17 observed by Acyl-Resin Assisted Capture (Acyl-RAC). We also demonstrate that expression of PRCD-R17C in vivo leads to protein mislocalization in the photoreceptor IS. Overall, we present additional evidence for the importance of palmitoylation transpiring at PRCD’s second amino acid, which sheds light on what is ensuing in the patients containing the R17C mutation and speculations backed by evidence of why an RP phenotype is observed in human patients.

## 2. Results

### 2.1. Mutation in the Polybasic Region (R17C) Significantly Destabilizes PRCD Protein

To determine the role of the polybasic region (PBR) in the PRCD protein, which is linked with retinitis pigmentosa (RP) in humans (Figure 1A), we cloned gBlock gene fragments of human wild-type and mutant PRCD constructs (PRCD-WT; PRCD-R17C; and PRCD-C2Y) tagged with Hemagglutinin (HA) in the C-terminal region under the control of a chicken β-actin promoter that is encoded by two restriction enzymes: Sal1 and BamH1 [5,12] (Figure 1A,B). Previous studies, including ours, demonstrated the strong association of PRCD to the membrane despite the lack of palmitoylation [8,10,11,12]. To better assess PRCD protein’s stability and expression when an R17C mutation is presented within the PBR, we used GFP protein as an internal loading control for PRCD that is driven independently by an internal ribosome entry site (IRES) (Figure 1B). PRCD-WT and mutant plasmid constructs were transiently transfected into hRPE1 cells to measure the stability of PRCD protein [12]. Immunoblot analysis reveals that an R17C mutation in the PBR affects protein stability by 45% (*p* < 0.001) compared with PRCD-WT. As demonstrated in our previous study, the stability of HA-tagged PRCD-C2Y protein is severely affected by >90% (Figure 1C,D). Altogether, our results show the mutation in the PBR (R17C) significantly affects protein stability despite having a cysteine at the 17th position, demonstrating that Arg17 is essential for PRCD protein stability. 

### 2.2. The Addition of a Cysteine in the PBR Does Not Prohibit Protein Lipidation and Membrane Association

As the mutation in the PBR (R17C) significantly affects protein stability, we assessed the lipidation status and membrane association of transiently transfected PRCD-R17C mutant protein in the hRPE1 cell line by acyl-RAC along with PRCD-WT, as described earlier [12]. Immunoblotting post-acyl-RAC revealed a robust acylation status in PRCD-R17C mutant protein compared to PRCD-WT (Figure 2A, lanes 10 and 15). As expected, in the control (-NH2OH), no proteins were seen in the elution fraction (Figure 2A, lanes 8 and 13). GFP protein served as an internal control and was not observed in the elution fraction with +NH2OH, suggesting PRCD-WT and R17C are both acylated despite having a mutation in the PBR and severe protein instability. A known palmitoylated protein, GAPDH is present in the elution fraction, serving as a positive control (Figure 2A, lanes 5, 10, and 15). The PBR in Ras and other proteins has been shown to enhance strong membrane association. Additionally, we previously demonstrated that PRCD strongly binds with membranes despite having a mutation in cysteine (C2Y). We speculate the transmembrane helices and adjacent PBR are likely playing a significant role in membrane anchoring. To determine how an R17C mutation in the PBR of PRCD affects membrane association, we performed membrane fractionation and evaluated it by immunoblot analysis. Our data shows a single mutation in the PBR does not affect the membrane association and the majority of protein remains strongly associated with the membrane (>90%), similar to WT and C2Y PRCD proteins (Figure 2B, lane 9; and C). A known membrane-binding protein, calnexin, which strongly associates with the membrane, served as a positive control (Figure 2B). GFP, serving as an internal control, is exclusively present in the soluble cytosolic fraction (Figure 2B). Overall, our data demonstrate the PRCD mutation in the PBR, linked with RP, neither affects lipidation nor disrupts membrane association.

### 2.3. Mutant Cysteine in the PBR Is Palmitoylated and Does Not Rescue Protein Stability despite Having a Strong Membrane Association

Although PRCD-R17C appears to be palmitoylated similar to WT PRCD, a limitation of acyl-RAC is being unable to determine which residue is being palmitoylated. Even though a mutation in the PBR could be detrimental to strong membrane binding, it is also possible that if this additional cysteine is acylated, that could contribute to membrane association in a different way. To determine if this could be a factor, we next created a double mutant that contains both a C2Y and an R17C mutation, PRCD-R17C/C2Y. Removing WT PRCD’s sole cysteine which is normally palmitoylated will allow us to use acyl-RAC to assess the palmitoylation status of this additional cysteine caused by an R17C mutation. Before proceeding to acyl-RAC, we sought to characterize this double mutant PRCD protein by first using palmitoylation prediction software (CSS-palm). This reveals a high probability of palmitoylation in the mutant cysteine at the 17th (Cys17) position (Figure 3A) along with the known palmitoylated Cys2. The amino acid sequence of the mutant construct made in the pCAG-IRES-eGFP vector is shown in Figure 3B. The transiently transfected PRCD double mutant shows a significant loss in protein stability compared to PRCD-WT (Figure 3C,D). However, the PRCD double mutant protein is comparatively more stable than the PRCD-C2Y protein. We showed in our earlier study that PRCD-C2Y mutation affects protein stability by 95% when compared to PRCD-WT. The PRCD double mutant protein (PRCD-R17C/C2Y) is 40% more stable when compared with C2Y (Figure 1D). As loss of palmitoylation of PRCD-C2Y has been shown to play a major role in the stability of the protein, this data suggests that Cys17 is likely being palmitoylated as predicted by CSS-Palm, stabilizing the double mutant protein. Together, these data demonstrate that while palmitoylation of PRCD is required for PRCD stability, the precise location of where lipidation transpires in PRCD could significantly attenuate protein stability if Cys2 is not palmitoylated. We next evaluated the acylation status of PRCD-R17C/C2Y by acyl-RAC. Immunoblot analysis demonstrates that the PRCD double mutant protein is acylated, as predicted, at the Cys17 (Figure 3E, lanes 5 and 10). GAPDH served as a palmitoylation positive control (Figure 3E, lanes 5 and 10). 

In order to determine the membrane association of PRCD-R17C/C2Y, we performed membrane fractionation as described earlier [12]. Transiently transfected PRCD and mutant proteins expressed in the hRPE1 cell line were used for membrane fractionation. Immunoblotting of fractionated samples revealed a strong membrane association of PRCD double mutant, similar to PRCD-WT (Figure 3F,G). As a control, we observed no changes in the solubility of GFP and membrane-bound protein calnexin fractionated in cytosolic and membrane fractions (Figure 3F). Together, our results indicate that the lack of endogenous palmitoylation and lipidation in the mutant cysteine (R17C), along with the transmembrane helices, is adequate for strong membrane association. 

### 2.4. Palmitoylation in the Cysteine-2 Position Is Indispensable for the Proper Localization of PRCD in hRPE1 Cells

The exclusive localization and strong membrane association of PRCD protein in the photoreceptor OS disc membrane is essential for OS maintenance [8,11,12]. Our previous study showed palmitoylation is essential for protein stability and proper localization [12,19]. As well, the specific site of palmitoylation at Cys2 is imperative. In the present study, we determine the localization of mutant PRCD proteins. The R17C mutation shows similar localization characteristics to PRCD-WT in hRPE1 cells where the majority of protein is localized in the cytosol (Figure 4A,B,E,F). Green represents Golgi marker, GM130 (Figure 4A,C,E,G), and green demonstrates the mitochondrial marker, HSP60 (Figure 4B,D,F,H). Palmitoylation deficient PRCD (C2Y) is aggregated and mislocalized in the subcellular compartments, particularly with HSP60, suggesting that palmitoylation deficient protein may be targeted to the mitochondria (Figure 4D). However, PRCD double mutant (PRCD-R17C/C2Y), which is acylated despite lacking its endogenous palmitoylation site (Cys2), is partially rescued from aggregation and localized to the subcellular compartment (Figure 4G,H). These results suggest that despite having palmitoylation in the PBR, endogenous palmitoylation at Cys2 is essential for proper localization. However, some PRCD-R17C proteins were localized similarly to PRCD-WT (Figure 4E,F). As demonstrated earlier, we treated the transiently transfected PRCD in hRPE1 cells with 2-bromopalmitate (2-BP), a known palmitoyl inhibitor, for 12 h after being transfected for 24 h [12,18]. Palmitoylation deficiency leads to the mislocalization of the PRCD protein into subcellular compartments. Similar to the C2Y mutant (Figure 4D), PRCD-WT, R17C, and C2Y/R17C proteins are aggregated and colocalized with HSP60 in the mitochondria (Figure 4J,N,P). This shows that loss of palmitoylation leads to PRCD protein aggregation and mislocalization to the mitochondria, likely leading to severe protein destabilization, as shown in Figure 1C and Figure 3C. Taken together, these results unequivocally demonstrate that endogenous palmitoylation at Cys2 is crucial for proper trafficking since C2Y/R17C protein, despite being palmitoylated in the mutated cysteine at the 17th position, remains mistrafficked to the subcellular compartments.

### 2.5. Subretinal Injection Followed by Electroporation in Murine Retina Unveils PRCD-R17C Mutant Protein Mislocalized to the Inner Segments

To investigate the localization of PRCD protein containing the mutation located within the PBR in the mouse retina, we performed subretinal injection at P0 of pups followed by electroporation of HA-tagged PRCD-WT, R17C, and R17C/C2Y plasmids (Figure 5A). As demonstrated in our earlier study, PRCD-C2Y protein lacking palmitoylation is mistargeted to the IS [12]. Immunohistochemistry of retinal cross-sections from PRCD-WT injected retina show the expression of PRCD-WT in the photoreceptor OS (Figure 5B) as previously reported [12]. GFP is used as a control, which is expressed in the same construct driven independently under an IRES, which is shown in the IS as demonstrated in previous studies (Figure 5B) [12,20,21]. The magnified image shows the localization of PRCD-WT protein in the OS and GFP in the IS (Figure 5B, enlarged).

Interestingly, the PBR mutant protein (R17C), which has an additional lipid modification, is localized to both the OS and IS of the photoreceptor cells (Figure 5G). Though we observed the PRCD-R17C localization in the OS, the majority of protein is mislocalized to the IS. In contrast, the PRCD double mutant (PRCD-R17C/C2Y) protein, which lacks palmitoylation at the normal residue (Cys2) and contains an R17C mutation that undergoes palmitoylation remains completely mistargeted to the IS (Figure 5J). However, a very small amount of PRCD-R17C/C2Y protein is observed in the OS, likely due to the lipidation of Cys17. Overall, these results demonstrate that despite the palmitoylation of Cys17 occurring in the PBR, exclusive endogenous palmitoylation at the Cys2 position is critical for efficient trafficking to the OS. Though the stability of the PBR mutation (R17C) is significantly improved when compared to the C2Y mutant protein (Figure 1D), it does not traffic efficiently to the OS and remains in the IS.

## 3. Discussion

In the current study, our lab sought to characterize the localization, acylation status, membrane association, and overall stability of PRCD containing an R17C mutation linked with RP in humans. The goal of our study was to understand the molecular determinants responsible for the strong disc membrane association of the PRCD protein, and its role in maintaining photoreceptor OS structure and function. As we demonstrated in our previous study, palmitoylation is not a primary determinant for membrane anchoring. We tested the role of a specific amino acid in the PBR in PRCD protein, where multiple mutations linked with RP are found, including Arg17Cys [5,15]. To understand the role of Arg17 in the PBR, we asked two questions: (1) does Arg17 of the PBR have a role in membrane anchoring, and (2) whether the mutant cysteine in the 17th position (R17C) forms a disulfide bond with the endogenous cysteine at the second position which could lead to protein de-stability and mistrafficking to subcellular compartments and ultimately cause the substitution of additional lipidation that is linked with RP transpiring in humans [5]. Our studies clearly demonstrate that mutation in the PBR (R17C) significantly affects protein stability despite being strongly membrane-associated. Most interestingly, the mutant cysteine at the 17th position is acylated, similar to the endogenous cysteine at the second position [12]. 

Protein acylation is a key regulator of many cellular functions, including protein trafficking, activity, protein-protein interaction, and membrane anchoring. We have demonstrated in an earlier study that palmitoylation deficiency in PRCD leads to severe protein destabilization and mis-trafficking to the subcellular compartments in the photoreceptor IS [8,12]. In rhodopsin, palmitoylation deficiency leads to visual impairment and light-induced photoreceptor degeneration [8,22,23,24]. Furthermore, in small GTPase Ras protein, there are multiple lipidations in the C-terminal region, along with the PBR, that are essential for efficient protein trafficking to subcellular compartments and strong membrane anchoring [25,26,27]. PRCD is predicted to have several structural domains, such as in the PBR region, as described in Figure 1 containing mutations linked with RP in humans [5,15]. In our earlier studies, we show despite lacking palmitoylation PRCD remains strongly associated with the membrane. This is likely supported by the highly conserved N-terminal transmembrane helices, followed by the PBR, which potentiate the strong membrane binding [12]. Our data from the present study reveals the PBR may not be necessary for such anchoring. In addition, our results show that the mutant cysteine in the PBR is potentially acylated by the Acyl-RAC approach in an in vitro cell culture system. Despite being acylated and showing strong membrane association, the stability of the PRCD-R17C mutant protein is significantly affected. 

Our data reveals that there is a possibility that the mutant cysteine in the PBR region does not form a disulfide linkage formation between the endogenous cysteine at the 2nd position. Mutation in the sole cysteine leads to severe protein de-stability and mistrafficking to subcellular compartments. In contrast, the R17C mutation does not cause severe loss of protein and is distributed similarly to wild-type PRCD. To further understand the strong membrane association, we deleted endogenous palmitoylation (C2Y) but retained the PRCD-R17C mutation in the PBR. Despite being acylated at the 17th position, the majority of PRCD-R17C/C2Y protein (~60%) is severely destabilized, although there is a strong membrane association. Taken together, this information supports our speculation that the highly conserved transmembrane helices (aa 3–15) could be a primary determinant for PRCD membrane association. Interestingly, a mutation in the endogenous cysteine (C2Y) linked with RP is aggregated and mis-trafficked to the sub-cellular compartments, particularly to the mitochondria where we observe colocalization with HSP60, a mitochondrial marker in the hRPE1 cell line. As indicated, palmitoylation of the second amino acid is crucial for trafficking to the OS with an unknown mechanism and deficiency could lead to defects in protein folding, which likely causes mis-trafficking to the subcellular compartments where it is destabilized rapidly by proteolytic degradation [12]. 

Our in vivo subretinal injection studies show a clear discrepancy in proper localization, where the R17C protein is mislocalized to the IS. However, with the loss of endogenous palmitoylation (PRCD-C2Y), lipidation in the PBR alone does not rescue proper localization and mutant PRCD remains ensnared in the IS. Taken together, we suggest the PBR likely supports membrane anchoring to an extent and additional lipidation does not support the trafficking to the OS, likely due to improper folding and aggregation to the IS. Based on these, despite being lipidated in the PBR, additional cellular stress could result from mistrafficking to the IS, which ultimately leads to protein destabilization and eventually photoreceptor death. To elucidate the effects of additional lipidation by observing the mutant cysteine 17 in PRCD (Cys2 and mCys17), our findings show that despite being additionally acylated in order for proper transport from the photoreceptor IS to OS to transpire endogenous acylation is essential. Any defects in this led to protein unstableness and caused additional cellular stress. Similar to the most common mutation in PRCD (C2Y), a mutation in the PBR affects trafficking to the OS which could lead to the disorganization of OS disc membranes [7,8,11,12]. Furthermore, our data show that PRCD’s strong membrane association is likely due to transmembrane helices and the PBR may contribute additional support for the strong photoreceptor disc membrane association but is not the primary anchor. However, the precise role of PRCD targeting with and without acylation needs to be clarified further. Endogenous palmitoylation at the Cys2 position transpires at the Golgi post-packaging through the endoplasmic reticulum (ER), but acylation of PRCD-R17C/C2Y could ensue at the ER because we demonstrate it is not entirely targeted to the Golgi. Consequently, we can look at the heterozygous PRCD-R17C mutation found in patients to hypothesize that since the mutation is relatively stable and partially trafficked to the photoreceptor OS in vivo, this may be why a patient of a subpopulation of patients with the mutation show a phenotype consistent with RP. It is important to reveal the crystal structure of PRCD protein to understand the mechanistic role of multiple structural domains in PRCD to discover its precise role in the OS and strong disc membrane association. Therefore, the present study is significant because the mutation in the PBR likely alters PRCD’s confirmation due to additional acylation which could lead to a defect in protein trafficking and stability linked with blinding diseases.

## 4. Materials and Methods

### 4.1. Reagents and Antibodies 

For Acyl-RAC purification the following chemicals were used: Lysis Buffer (25 mM HEPES, pH 7.5 [Gibco, Waltham, MA, USA], 25 mM NaCl [Invitrogen, Waltham, MA, USA], 1 mM EDTA [Sigma-Aldrich, Burlington, MA, USA], protease and phosphatase inhibitors), Blocking Buffer (100 mM HEPES, pH 7.5 [Gibco, Waltham, MA, USA], 1 mM EDTA [Sigma-Aldrich, Burlington, MA, USA], 2.5% SDS [Sigma-Aldrich, Burlington, MA, USA], 0.1% S-methyl methanethiosulfonate [Sigma-Aldrich, Burlington, MA, USA]), Binding Buffer (100 mM HEPES, pH 7.5 [Gibco, Waltham, MA, USA], 1 mM EDTA [Sigma-Aldrich, Burlington, MA, USA], 1% SDS [Sigma-Aldrich, Burlington, MA, USA]), 2 M Hydroxylamine, pH 7.5 [Thermo Scientific, Waltham, MA, USA], 2 M NaCl [Invitrogen, Waltham, MA, USA] and Agarose S3 High Capacity Acyl-8;’ Capture Resin [Nanocs, New York, NY, USA]. When cloning and cell culture experiments progressed, the chemicals used were: DMEM/F12 medium (10% Fetal Bovine Serum [Cytiva, Marlborough, MA, USA], 1% penicillin-streptomycin [Fisher, Waltham, MA, USA]), Trans-LT1 Transfection Reagent [Mirus, Madison, WI, USA], Dulbecco’s phosphate-buffered saline (1 × DPBS [Corning, Corning, NY, USA] and 0.25% Trypsin-EDTA [Gibco, Waltham, MA, USA], dimethyl sulfoxide (DMSO) [Sigma-Aldrich, Burlington, MA, USA], 2-bromopalmitate [Sigma-Aldrich, Burlington, MA, USA]. For immunohistochemistry and immunocytochemistry, the reagents used were: 4% Paraformaldehyde (16% PFA, [Electron Microscopy Sciences, Hatfield, PA, USA]) and 5% Goat Serum (Normal Goat Serum [Millipore, Burlington, MA, USA], 0.5% Triton-X-100 [Sigma-Aldrich, Burlington, MA, USA], 0.05% Sodium azide [MP, Santa Ana, CA, USA]). Measurement of protein levels transpired with 1× phosphate-buffered saline, PierceTM Protease Inhibitor Mini Tablets, EDTA-free [Fisher, Waltham, MA, USA], 1× phosphate buffered saline Tween-20 [Fisher, Waltham, MA, USA], InterceptTM Antibody Diluent T20 PBS [Li-Cor, Lincoln, NE, USA], Intercept^®^ Blocking Buffer PBS [Li-Cor, Lincoln, NE, USA]. Subretinal injection requirements included 0.1% fluorescein in PBS (AK-FLUOR) [Alcon, Fort Worth, TX, USA]. 

### 4.2. Animals

All handling, care, and experimental procedures involving animals were performed in agreement with National Institutes of Health guidelines. The Institutional Animal Care and Use Committee (IACUC) at West Virginia University approved the protocol used. 

### 4.3. Cloning and Cell Transfection

hTERT-immortalized pigment epithelial (hRPE1) cells were used and transfected with constructs encoding human PRCD, either wild type (PRCD-WT), PRCD-C2Y, PRCD-R17C or PRCD-R17C/C2Y mutants, with C-terminal HA epitope-tagged gBlocks gene fragments (200 ng). These gene fragments were cloned into a pCAG-IRES-eGFP vector, as described in our earlier study, where PRCD and HA are under a chicken actin promoter [12]. Endogenous expression of eGFP, used as an internal control, is driven independently by IRES element. The hRPE1 cells were maintained in DMEM/F12 medium enriched with 10% FBS and 1% penicillin-streptomycin in a sterile incubator at 37 °C with 5% CO_2_. The cells were cultured on 100 mm circular dishes and split when 70–90% confluent onto 6-well plates. For the transfection of hRPE1 Bio Mirus’s Trans-LT1 was used, and the manufacturer’s protocol was followed. 48 h post-transfection, cells were collected and stored for future analysis as described below. 

### 4.4. Immunoblotting

Lysates of hRPE1 cells transiently transfected with PRCD protein expressing constructs were collected as previously explained, and homogenized with 1 × PBS containing protease and phosphatase inhibitors (Pierce) [12]. Protein concentration was analyzed by the standard BCA method with a NanoDrop spectrophotometer (ND-1000, Thermo Scientific). A 15% SDS-PAGE was used to run the samples at equal concentrations and transferred to an Immobilon-FL membrane. After transfer, the membrane was blocked with Intercept^®^ Blocking Buffer PBS (Licor) for one hour followed by primary antibody incubation for two hours at room temperature. Antibodies from Table 1 were diluted in a 1:1 mixture of 1 × PBST (Tween-20) with Intercept™ Antibody Diluent T20 PBS. Following incubation were three 1 × PBST washes for five minutes each and then a 30-min incubation with secondary antibodies diluted in 1 × PBST per concentrations mentioned in Table 1. After three washes for five minutes each with 1 × PBST, the blots were scanned, and densities were measured with an Odyssey Infrared Imaging System (LI-COR Biosciences) per the manufacturer’s guidelines.

### 4.5. Acyl-RAC Purification (Palmitoylation Assay)

The palmitoyl modification of PRCD was assessed using Acyl-RAC, as previously described [12,28]. Transiently transfected hRPE1 cells were collected as described in the following. Each sample’s pellet was resuspended in 300 µL of ice-cold lysis and sonicated at 12 psi for three rounds of three seconds, being returned to the ice for 30 s between each round. Proteins were solubilized with 1% Triton-X-100 by incubating on a nutator at 4 °C for 20 min. Post-incubation, samples were centrifuged at 200× *g* for three minutes and the supernatant of each sample was split equally into 3 tubes, 100 µL in each. To block the free cysteines, 300 µL of blocking buffer containing Methyl-methane Thio-sulfonate (MMTS) was added to each 100 µL aliquot (3 aliquots per sample) and incubated at 40 °C for 16 min, while being vortexed every 2 min. Ice-cold acetone was added to precipitate proteins during a 20-min incubation at −20 °C. Samples were centrifuged at 20,000× *g* for 15 min at 4 °C, then the supernatant was removed and the pellets were washed with 70% acetone four times. Post wash, the pellets were air dried for 30 min at room temperature, resuspended with 200 µL of binding buffer, and sonicated to be solubilized. Each sample’s three aliquots were pooled together (600 µL total per sample) in one tube. The assay was performed for each sample treating with and without hydroxylamine (“+HAM” and “−HAM”) with 75 µL of Agarose S3 High Capacity Acyl-RAC Capture Resin (Nancos) beads, prewashed with binding buffer. 45 µL of sterile water was added in the “−HA” tube and 45 µL of 2 M Hydroxylamine (NH2OH, HA), pH 7.5 was added in the “+HA” tube, respectively. 250 µL of each pooled sample was added to the “−HA” and “+HA” tubes and all samples were incubated on a nutator at room temperature for two hours. After the incubation, the beads were then washed four times with binding buffer. The bound fractions were eluted with 1X Lamellae sample buffer containing 10 mM DTT by boiling for five minutes and then loaded into an SDS Page Gel.

### 4.6. Subcellular Protein Fractionation

Two protocols were used for cellular fractionation to complement and verify the result. Subcellular Protein Fractionation Kit for Cultured Cells (Fisher Scientific) was used and the protocol was executed through Step 5 to obtain the “membrane extract”. To verify, we used the same amount of collected cells per sample (3 wells of a 6-well plate) and started by suspending them in 300 µL of isotonic buffer (1 × PBS) containing phosphatase and protease inhibitors (Pierce). The cells were sonicated three times for three seconds each at 12 amps. Between each sonication, the samples were kept on ice for 30 s. The samples were then centrifuged at 500× *g* for three minutes at 4 °C. After collecting the total fraction, the remaining supernatant of each sample was ultra-centrifuged at 45,000× *g* for 15 min at 4 °C. 75 µL of the supernatant was collected and each was labeled as the appropriate “Soluble Fraction”. The remaining supernatant was aspirated. Each pellet was resuspended in 75 µL of isotonic buffer with inhibitors and labeled as the “Membrane Fraction’’. 5× sample buffer was added to each final fractions Total, Soluble, and Membrane tubes and boiled for five minutes before loading into an SDS-PAGE gel followed by immunoblotting.

### 4.7. Immunocytochemistry

hRPE1 cells were cultured on sterile circular glass coverslips and transfected with PRCD constructs as previously described [12]. After 48-h transfection, the media was aspirated and each well was washed with 1 × PBS for one minute, followed by fixation in 4% paraformaldehyde (PFA) for ten minutes, and then rinsed for 30 s with 1 × PBS. Permeabilization was conducted with cold methanol for five minutes before three two-minute washes with ice-cold 1 × PBS. Cells were blocked with 5% goat serum for one hour at room temperature. Briefly, slides were washed with 1 × PBS before the addition of primary antibodies overnight at 4 °C. The following day, each well was washed with 1 × PBS Triton-x-100 (PBST) three times for five-minutes a piece. A secondary antibody was added for one hour at room temperature. Followed by three five-minute washes with 1 × PBS. Each slide was then mounted with ProLong™ Gold antifade reagent.

### 4.8. Immunohistochemistry

After eyecups were sectioned at a thickness of 16 µm onto a Superfrost Plus Microscope Slide (Fisher) a hydrophobic barrier was drawn with a PAP pen around the desired sections. The sections were washed with 1 × PBS three times for five minutes each and then were incubated with blocking buffer (1 × PBS, 10% normal goat serum, 0.5% Triton-X, 0.05% sodium azide) for one hour at room temperature. The blocking buffer was aspirated, and the sections were incubated overnight at 4 °C with the primary antibodies (Table 1) of choice, diluted appropriately in antibody dilution buffer (ADB) (1 × PBS, 5% normal goat serum, 0.5% Triton-X, 0.05% sodium azide). The next morning the primary antibody was aspirated, and the sections were washed with 1 × PBST (0.1% Triton-X-100) for 15 min once and twice with 1 × PBS for 15 min. The corresponding secondary antibody (Table 1) was diluted in ICC ADB and the antibodies were incubated in the sections for one hour at room temperature. The sections were then washed twice for 15 min with 1 × PBST and then repeated once with a 1 × PBS wash. After the final wash, the slides were aspirated and set out to dry for a minute. ProLong™ Gold antifade reagent (Invitrogen) was applied to the slides prior to being mounted with coverslips and left to dry for at least 24 h before imaging.

### 4.9. Subretinal Injection

Plasmid DNA of PRCD-WT, PRCD-R17C, and PRCD-R17C/C2Y was purified and diluted to a concentration of 5 μg/µL along with 0.1% fluorescein in 1 × PBS (100 mg mL^−1^ AK-FLUOR, Alcon, Fort Worth, TX). On postnatal days zero to three, approximately 24 to 30 pups (4 litters) were injected for each construct. An average of 60–70% of the pups were shown to have positive injections with the range of variables between 30% to 90% expression of injected PRCD proteins. We evaluated the mislocalization of PRCD mutant proteins compared with wild-type PRCD showing >90% of mutant protein stuck in the inner segment of photoreceptor cells. CD1 (Charles River) pups were anesthetized by hypothermic conditioning obtained by keeping them on ice for seven to ten minutes. Using a dissecting microscope, the future eyelids were slit open to give access to the developing eye. One eye was injected into the PRCD mutant construct in the subretinal space and the other contralateral eye was injected with 1 × PBS to serve as a control. A 30G ½ needle was used to make a hole alongside the pupil so a blunt end needle (33 gauge, 6 pk, 10 mm, 45° angle [Hamilton—Cat #7803-05]) with 0.5 µL of the DNA/fluorescein solution could be injected into the subretinal area located behind the lens. Following injection, electroporation was conducted with tweezer-like paddles (BTX model 520, 7-mm diameter) for five pulses at 80 v for 50 ms durations with 950 ms rests in between. Eyes were collected on postnatal day 21, followed by a check for GFP fluorescence activity in the eye to see if the plasmid DNA was taken up by the developing photoreceptors, which was found to be the case in 60–70% of the injected eyes. Retinal sectioning and immunohistochemistry were conducted as previously described [12].

### 4.10. Statistical Analysis

Data are expressed as means ± S.E. unless otherwise indicated. The differences between wildtype control (PRCD-WT) and Mutants PRCD (PRCD-C2Y, PRCD-R17C, PRCD-R17C/C2Y) were analyzed with a two-tailed student test-test (online version accessed on 26 July 2021, http://www.usablestats.com/calcs/2samplet).

## Figures and Tables

**Figure 1 ijms-23-10802-f001:**
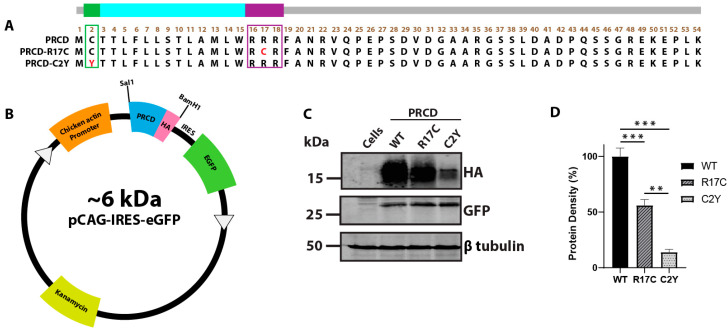
Arginine 17 in PRCD is essential for protein stability. (**A**) the amino acid sequence of PRCD protein depicting PRCD-WT, PRCD-R17C, and PRCD-C2Y. The green area indicates the site of palmitoylation at the second cysteine residue, and the most common mutation “Cys2Tyr” is indicated in red. The purple box depicts the polybasic region from the 16–18 aa and the R17C mutation is shown in red. The transmembrane helix is coded blue from the third to the fifteenth aa. (**B**) cloning cassette scheme demonstrating the PRCD wildtype and mutants tagged with HA (hemagglutinin) expressed under a chicken actin promoter. In addition, eGFP expressed independently under IRES (Internal Ribosome Entry Site) serves as an internal control. (**C**) immunoblot of PRCD-WT, R17C, and C2Y transiently transfected in hRPE1 cells. GFP serves as an internal control, with β tubulin being a loading control. (**D**) quantification of PRCD proteins identified in C (*n* = 3). The expression of PRCD was normalized to the internal GFP control (**, *p* < 0.005 and ***, *p* < 0.0005) based on a one-tailed Student *t*-test.

**Figure 2 ijms-23-10802-f002:**
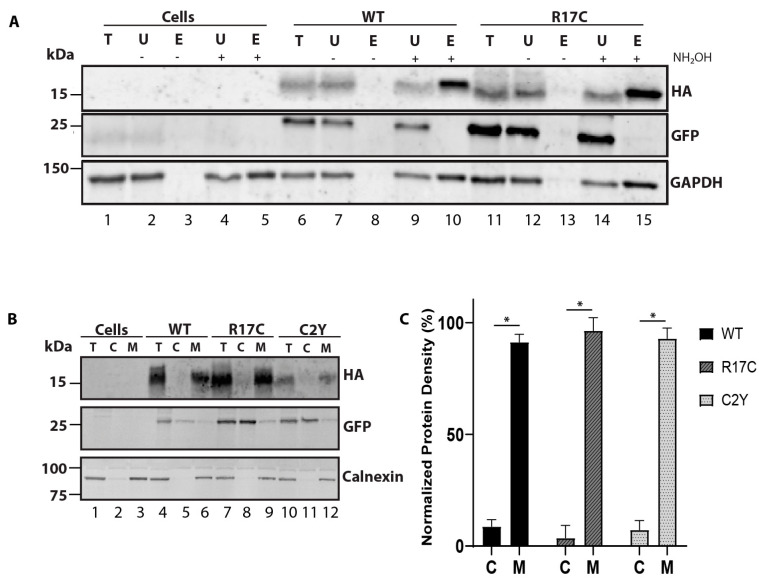
Mutation in the PBR region of PRCD does not affect protein palmitoylation and membrane association. (**A**) to assess the palmitoylation status of PRCD-R17C, the Acyl-RAC method was conducted with protein transiently expressed in hRPE1 cells. Immunoblots show both PRCD-WT and R17C are palmitoylated after treating with hydroxylamine (+NH2OH) (compare lanes 8 and 10 (WT); 13 and 15 (R17C)), whereas -NH2OH extracts were treated with vehicle control of water (*n* = 3). Total (T), unbound (U), elute (E) protein fractions, -E (untreated with NH2OH), +E (treated with NH2OH). GFP serves as a negative control, and GAPDH serves as a known palmitoylated positive control. Non-transfected cells (lanes 1–5) are used as a control which does not express -tagged PRCD proteins. (**B**) isotonic protein extraction examining the associations of proteins with membranes. Fraction labeling is denoted by: total (T), cytosolic (C), and membrane (M). HA, PRCD protein tagged with HA-tag, GFP, a known cytosolic protein independently expressed under IRES in the same plasmid construct, and calnexin, a known membrane-bound protein, were used as controls. Each PRCD mutant protein shows strong membrane binding. (**C**) Protein density of each fraction was normalized to protein density found in the total protein fractions (*n* = 3) with each mutant being significantly (* *p* < 0.05) bound to the membrane.

**Figure 3 ijms-23-10802-f003:**
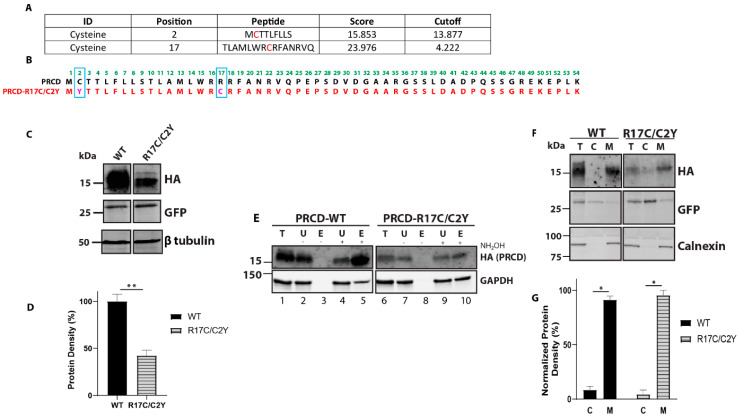
Cysteine residue of mutant PRCD-R17C/C2Y is palmitoylated and strongly binds with membranes despite the severe protein instability. (**A**) CSS-Palm prediction data showing PRCD-R17C is predicted to be lipidated in both Cys2 and Cys17 positions accessed on 30 November 2021 (http://csspalm.biocuckoo.org). (**B**) Cloning of PRCD double mutant, PRCD-R17C/C2Y, marked in pink with blue boxed along with PRCD-WT. (**C**) immunoblot showing the stability of proteins: PRCD-WT and PRCD-R17C/C2Y. GFP was used as an internal loading control, with β-tubulin as an external loading control. (**D**) quantification of protein stability from panel C (*n* = 3, **, *p* < 0.005). (**E**) Acyl-RAC of PRCD-WT and PRCD-R17C/C2Y show mutant cysteine in the 17th position is palmitoylated. GAPDH serves as a positive palmitoylation control. (**F**) subcellular fractionation demonstrates both wildtype and mutant PRCD are primarily membrane (M) bound and absent from cytosol (C). As controls, GFP was used as a cytosolic marker and calnexin as a membrane marker. (**G**) quantification of protein localization to either cytosolic fraction or membrane fraction of PRCD-WT and PRCD-R17C/C2Y (*n* = 3, *, *p* < 0.05).

**Figure 4 ijms-23-10802-f004:**
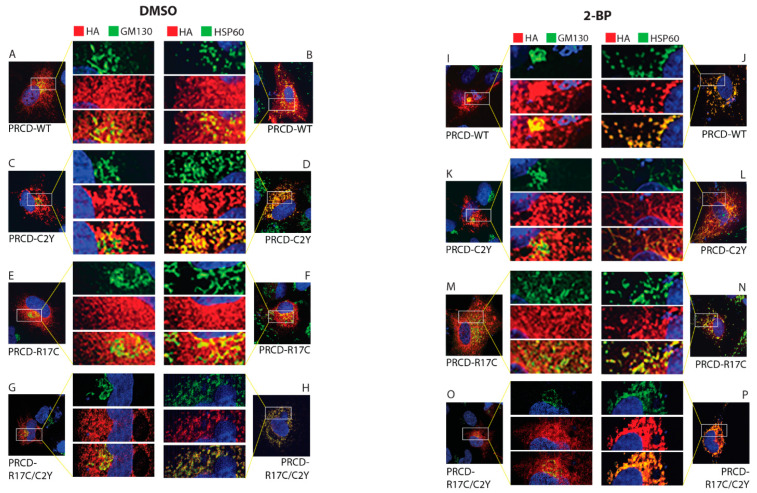
Palmitoylation of cysteine in the second amino acid position is essential for proper localization. Immunocytochemistry of transiently transfected hRPE1 cells with PRCD-WT, and PRCD-R17C display similar localization (**A**,**E**; **B**,**F**). Endogenous palmitoylation lacking PRCD-C2Y, and double mutant PRCD-R17C/C2Y showing aggregation mislocalization in the subcellular compartment likely localized with HSP60, a mitochondrial marker (**C**,**G**; **D**,**H**). PRCD’s (HA, red) localization in regards to the Golgi complex (GM130, green) and Mitochondria (HSP60, green). Nuclei are stained blue with DAPI. A-H, hRPE1cells treated with control vehicle, DMSO. (**I**–**P**) hRPE1 cells treated with 150 μM 2-BP, a palmitoyl inhibitor, for 24 h post-transfection show mislocalization with HSP60. The middle panel show enlarged images.

**Figure 5 ijms-23-10802-f005:**
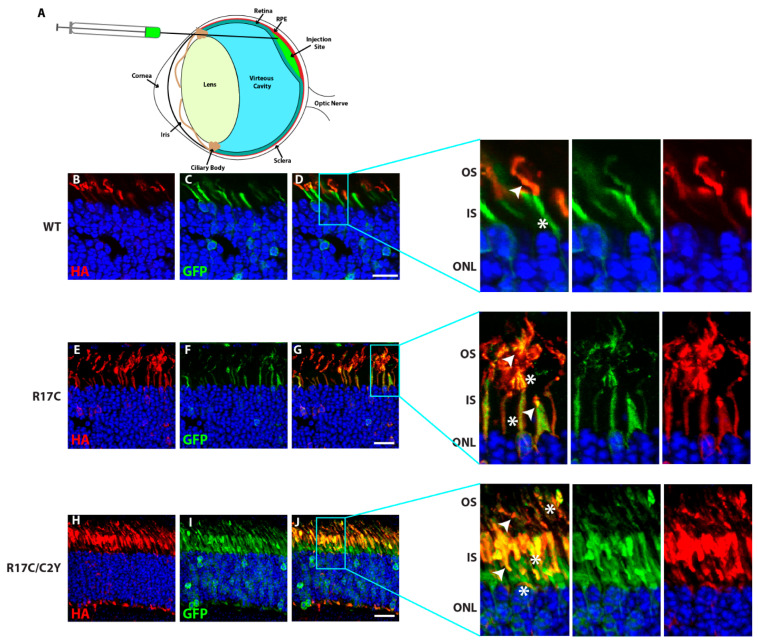
Additional palmitoyl lipid modification in PRCD does not rescue its localization to the photoreceptor OS and remains severely mislocalized in the IS. (**A**) Illustration depicting the delivery of various PRCD constructs into the subretinal space. (**B**–**J**) Subretinal injection followed by electroporation of HA-tagged PRCD-WT (**B**–**D**), R17C (**E**–**G**), and R17C/C2Y (**H**–**J**) double mutants in CD1 mouse retina. Localization of PRCD-WT in the photoreceptor OS (B-D and expanded images in the right panel highlighted). In contrast, PRCD-R17C is localized to both the OS and IS (**E**–**G**) and expanded image in the right panel), and R17C/C2Y is highly mislocalized to the IS. The white arrow depicts HA localization, and the white asterisk shows GFP location in the photoreceptor. GFP (green) serves as a transfection control localized to the photoreceptor IS as described in earlier studies. The right panels are magnified to show the localization patterns of PRCD proteins. Wild-type PRCD is exclusively expressed in photoreceptor OS, and R17C shows both mislocalizations to the IS and normal localization in OS, whereas the majority of double mutant PRCD-C2Y/R17C protein is mislocalized to the inner segment. Scale bar 20 µm.

**Table 1 ijms-23-10802-t001:** Antibody List.

Antibody	Dilution and Application		Animal Raised In	Source	Identifiers
	WB	ICC/IHC			
Anti-HA High Affinity	1:2000	1:1000	Rat	Sigma-Aldrich	Cat# 11867423001 RRID: AB_390918
GFP tag	1:2000	1:1000	Mouse	Protein Tech	Cat# 66002-1-Ig RRID: AB_11182611
Anti-β- Tubulin	1:3000	N/A	Mouse	Sigma-Aldrich	Cat# T5201 RRID: AB_609915
GAPDH	1:10,000	N/A	Mouse	Protein Tech	Cat# 10494-1-AP RRID: AB_2263076
Anti- Calnexin (TO-5)	1:200	N/A	Mouse	Santa Cruz Biotechnology	Cat# sc-80645 RRID: AB_1119919
Anti-GM130	N/A	1:200	Mouse	BD Sciences	Cat# 610822 RRID: AB_398142
HSP60	N/A	1:500	Mouse	Protein Tech	Cat# 66041-1-Ig RRID: AB_11041709
Alexa Fluor™ 568 goat anti-rat IgG	1:50,000	1:500	Goat	Invitrogen	Cat# A11077 RRID: AB_2534121
Alexa Fluor™ 680 goat anti-rat IgG	1:50,000	1:500	Goat	Invitrogen	Cat# A21096 RRID: AB_2535750
Alexa Fluor™ 680 goat anti-mouse IgG	1:50,000	1:500	Goat	Invitrogen	Cat# A21058 RRID: AB_2535724
IRDye^®^ 800CW Goat anti-Mouse	1:50,000	1:500	Goat	Li-Cor	Cat# 926-32210 RRID: AB621842
4′,6-diamidino-2-phenylindole (DAPI)	N/A	1:1000	N/A	Thermo Fisher	Cat# D1306, RRID: AB_2629482

## Data Availability

Not applicable.

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
