# Peer review of "R17C Mutation in Photoreceptor Disc-Specific Protein, PRCD, Results in Additional Lipidation Altering Protein Stability and Subcellular Localization"

_ijms, 2022, doi:10.3390/ijms231810802_

Round 1
Reviewer 1 Report
In addition to the C2Y mutation, which has been frequently observed among Pred gene variants, this paper describes the C17R mutation, whose pathogenicity has been questioned, and shows that the C17R mutation can also cause RP by clarifying the biochemical properties of proteins with the C17R mutation. The paper is well written and the logic is clear. It is of scientific value in examining how a single gene variant can alter the proteins it introduces and how this can affect the establishment of a disease.
Author Response
We appreciate the reviewer’s comments and agree with the reviewer, as R17C pathogenicity is not clearly understood due to the frequency of mutation. However, we chose to work with the R17C mutation as it is located in the polybasic region (PBR). As PRCD strongly associates with the photoreceptor disc membrane, the strong membrane association is not clearly understood. As we previously demonstrated, palmitoylation deficiency doesn’t affect membrane anchoring, so we believe that the PBR may be playing a role for membrane anchoring. The present study shows two important observations: 1) despite having a mutation in the PBR, the mutant protein remains strongly associated with the membrane, and 2) the mutant cysteine in the 17th position undergoes additional lipid modification neither rescuing protein stability nor trafficking to the photoreceptor outer segment.
Reviewer 2 Report
The authors provide an interesting manuscript on the mutation in the PRCD gene and its effects on protein stability and subcellular localization. The manuscript is clear and they do a nice job presenting complex findings. The authors should add additional details as noted below, including information on their retinal images. Additional specific comments are included below.
# Author needs to rethink including the RP in the title, it might mislead the readers. Since none of the experiments were done related to RP condition.
# In figure 5 authors demonstrate the PRCD rescue and localization in IS vs OS. It would be consistent to show the WT image (5B) with more photoreceptors, similar to what is seen in the double mutation image (5H).
#, with Figure 5, did authors Costain the mouse retina with rods or cones-specific markers to determine the cell-specific localization defects?
# did authors notice any change in the localization differences in dorsal vs ventral retina? It is known that there are differences in photoreceptors count. Please describe if there are any differences.
# line 205 “period” needs to be included.
# line 209 authors describe the mitochondrial marker HSP60 in red, while the figure 4 legend and image are displayed as green. Please correct it accordingly.
# authors need to add the details if both eyes were injected in the same pups. In total how many pups were injected and analyzed? Can authors provide the statistics for cell mislocalization?
# Please also include the information for the thickness of retinal sections in the methods section.
Author Response
Reviewer 2:
The authors provide an interesting manuscript on the mutation in the PRCD gene and its effects on protein stability and subcellular localization. The manuscript is clear, and they do a nice job presenting complex findings. The authors should add additional details as noted below, including information on their retinal images. Additional specific comments are included below.
- Author needs to rethink including the RP in the title, it might mislead the readers. Since none of the experiments were done related to RP condition.
Thank you to the reviewer for the concerns regarding including “RP” in the title. We changed the title as per the author suggestion to “R17C mutation in photoreceptor disc-specific protein, PRCD, results in additional lipidation altering protein stability and subcellular localization”
- In figure 5 authors demonstrate the PRCD rescue and localization in IS vs OS. It would be consistent to show the WT image (5B) with more photoreceptors, similar to what is seen in the double mutation image (5H).
Thank you to the reviewer for this concern. We attempted injecting WT PRCD into retina at different ages (P0, P1, and P2) on several occasions. Unfortunately, over expression of wild type PRCD is toxic to the photoreceptor cells and expresses very little. Currently, we are in the process of studying PRCDs role using transgenic animals expressing wild type and mutant (C2Y) PRCD. We noticed, injecting mutant PRCD in the subretinal space doesn’t affect its expression. We appreciate again the reviewer for pointing this out.
- With Figure 5, did authors Costain the mouse retina with rods or cones-specific markers to determine the cell-specific localization defects?
Thank you, we and others have previously described PRCD expression in both rods and cones. Please refer to the following paper (doi: 10.1074/jbc.M116.742767. Epub 2016 Sep 9).
- Did authors notice any change in the localization differences in dorsal vs ventral retina? It is known that there are differences in photoreceptors count. Please describe if there are any differences.
We appreciate the question from the reviewer. Our immunolocalization studies did not show any changes or differences in dorsal/ventral retina. However, as we inject PRCD constructs in the subretinal space, the wild type PRCD always express less than the mutant construct. At present, we are exploring this using our transgenic animal model to elucidate why over expressing PRCD affects retinal function and survival. Also, we are exploring PRCD mediated proteasome insufficiency leading to retinal degeneration.
- Line 205 “period” needs to be included.
A “period” has been added tot line 205.
- Line 209 authors describe the mitochondrial marker HSP60 in red, while the figure 4 legend and image are displayed as green. Please correct it accordingly.
We apologize for the error and thank the reviewer for critically reading our manuscript. We corrected HSP60 to be indicated in green and the manuscript has been updated.
- Authors need to add the details if both eyes were injected in the same pups. In total how many pups were injected and analyzed? Can authors provide the statistics for cell mislocalization?
In each pup, we injected one eye with the PRCD construct and the contralateral eye was injected with 1X PBS. The contralateral eye injected with 1X PBS served as a control and was collected at the same time as the injected with the PRCD construct. In total, 24 to 30 pups (4 litters) were injected for each construct. The average of 60-70% of the pups were shown positive injections with the range of variables between 30% to 90% expression of injected PRCD proteins. We evaluated the mislocalization of PRCD mutant proteins compared with wild type PRCD which shows >90% of R17C/C2Y mutant protein and >60% of R17C mutant protein is stuck in the inner segment of photoreceptor cells. This information added in line 495-498.
- Please also include the information for the thickness of retinal sections in the methods section.
Thank you, we included the thickness of retinal sections in the method, which is 16 µm.
Reviewer 3 Report
In this paper, Myers et al. analyzed an Arg17Cys mutation in the PRCD gene associated with retinitis pigmentosa. The authors characterized the stability, acylation status, membrane association, and localization of PRCD containing an Arg17Cys mutation. The overall study design, methods, and the other sections of the manuscript, including figures and tables, are well organized and illustrate the facts discussed in the text. The interpretation is consistent with the findings. However, there are some questions and concerns:
1. Line 55-57: Does this polymorphic variation (R17C) listed in the dbSNP or any other database? If yes, please report the minor allele frequency in the Introduction part.
2. Line 58-59: There is no reference for the Indian-origin patient with a heterozygous R17C mutation. Are there any other published data showing RP patients with this mutation?
3. The authors have characterized the role of Arg17 amino acid in PRCD however this variation might not be enough to cause RP phenotype as it exists in the general population without RP. The authors should mention this in the Discussion part.
Line 37-38> PRCD has been shown to be localized specifically… This sentence is not properly constructed, consider revising it.
Line 121> demonstrate>> demonstrated.
Line 131> our data demonstrates>> our data demonstrate.
Line 205-206> This sentence is not properly constructed, consider revising it. A period is missing.
Finally, a more tightly worded manuscript with the above-suggested corrections will make the paper a good read.
Author Response
Reviewer 3:
In this paper, Myers et al. analyzed an Arg17Cys mutation in the PRCD gene associated with retinitis pigmentosa. The authors characterized the stability, acylation status, membrane association, and localization of PRCD containing an Arg17Cys mutation. The overall study design, methods, and the other sections of the manuscript, including figures and tables, are well organized and illustrate the facts discussed in the text. The interpretation is consistent with the findings. However, there are some questions and concerns:
- Line 55-57: Does this polymorphic variation (R17C) listed in the dbSNP or any other database? If yes, please report the minor allele frequency in the Introduction part.
We are thankful to the reviewer for enquiring about the polymorphic variation of PRCD R17C in the dbSNP. We checked in the Global variome shared LOVD (https://databases.lovd.nl/shared/variants/0000796901#00016771) and found there are four reported entries, one of which was a homozygous mutation reported in 2014 (PMCID: PMC3945441). As mentioned, we included this in the introduction as highlighted.
- Line 58-59: There is no reference for the Indian-origin patient with a heterozygous R17C mutation. Are there any other published data showing RP patients with this mutation?
Thank you, the reference for the patient containing RP is included in the manuscript (Zangerl, et.al. (2006), which was referenced in the previous sentence. Also, we included another reference showing R17C mutation linked with RP patient (Hum Genet. 2014 March; 133(3): 331–345. doi:10.1007/s00439-013-1381-5).
- The authors have characterized the role of Arg17 amino acid in PRCD however this variation might not be enough to cause RP phenotype as it exists in the general population without RP. The authors should mention this in the Discussion part.
We agree, one of our points is the Arg17Cys mutation may not be sufficient to cause the RP phenotype with each patient containing the mutation. Towards the end of the Discussion portion of the manuscript, around line 353, we addressed this possibility.
- Line 37-38> PRCD has been shown to be localized specifically… This sentence is not properly constructed, consider revising it.
Thank you, we have revised the sentence by saying: “In photoreceptors, PRCD is synthesized in the IS, trafficked to the OS, and strongly associated in the disc membranes. Although the precise role of PRCD in the disc membranes remains unclear, our lab has recently demonstrated that PRCD plays a crucial role in regulating the packaging of rhodopsin into OS disc membranes (7-9).”
- Line 121> demonstrate>> demonstrated.
We have changed “demonstrate” to “demonstrated”.
- Line 131> our data demonstrates>> our data demonstrate.
We have changed “demonstrates” to “demonstrate”.
- Line 205-206> This sentence is not properly constructed, consider revising it. A period is missing.
We have added a period in line 205.
To the sentence found in line 205-206, we have made the following changes: “In the present study we determine the localization of mutant PRCD proteins. The R17C mutation shows similar localization characteristics to PRCD-WT in hRPE1 cells where the majority of protein is localized in the cytosol (Fig.4 A-B,E-F).”
Finally, a more tightly worded manuscript with the above-suggested corrections will make the paper a good read.